# Anchoring Apparatus of Long Head of the Biceps Tendon: Ultrasonographic Anatomy and Pathologic Conditions

**DOI:** 10.3390/diagnostics12030659

**Published:** 2022-03-08

**Authors:** Heng Xue, Stephen Bird, Ling Jiang, Jie Jiang, Ligang Cui

**Affiliations:** 1Department of Ultrasound, Peking University Third Hospital, Beijing 100191, China; xueheng1987@bjmu.edu.cn (H.X.); papayaling@163.com (L.J.); jiangjie_us@163.com (J.J.); 2Benson Radiology, Wayville, SA 5034, Australia; sjbird@ozemail.com.au

**Keywords:** ultrasonography, scanning technique, shoulder, long head of the biceps tendon, rotator cuff

## Abstract

The long head of the biceps tendon (LHBT) has been recognized as an important generator of anterior shoulder pain, causing a significant reduction in the shoulder flexion range. Various tendinous and ligamentous structures form the anchoring apparatus of the LHBT along its course to maintain its appropriate location during shoulder movements, including the coracohumeral ligament (CHL), superior glenohumeral ligament (SGHL), subscapularis (SSC) tendon and supraspinatus (SSP) tendon as well as the less recognized tendons of pectoralis major (PM), latissimus dorsi (LD) and teres major (TM). Lesions of this stabilizing apparatus may lead to an instability of the LHBT, resulting in pain at the anterior shoulder. Ultrasonography (US) has been increasingly used in the assessment of shoulder injuries, including the anchoring apparatus of the LHBT. An accurate diagnosis of these injuries is often challenging, given the complex anatomy and wide spectrum of pathologies. In this review article, US anatomy and common pathologic conditions that affect the anchoring apparatus of the LHBT are discussed, including biceps pulley lesions, adhesive capsulitis, chronic pathology of SSC and SSP tendons, tears in the PM tendon and injuries to the LD and TM. Knowledge of a normal anatomy, an appropriate scanning technique and US findings of common pathologic conditions are the keys to accurate diagnoses.

## 1. Introduction

Shoulder pain is a common complaint encountered in clinical practice. Beyond rotator cuff abnormalities, the long head of the biceps tendon (LHBT) has been recognized as an important generator of anterior shoulder pain, causing a significant reduction in the shoulder flexion range. Originating from the supraglenoid tubercle of the scapula, the LHBT first travels intra-articularly then turns 30 degrees over the humeral head [1,2]. Due to this anatomical characteristic, the LHBT is predisposed to medial dislocation or subluxation from the bicipital groove [2,3]. Various tendinous and ligamentous structures form the anchoring apparatus of the LHBT along its course to maintain its appropriate location during shoulder movements, including the coracohumeral ligament (CHL), superior glenohumeral ligament (SGHL), subscapularis (SSC) tendon and supraspinatus (SSP) tendon as well as the less recognized tendons of pectoralis major (PM), latissimus dorsi (LD) and teres major (TM) [2,4,5]. Lesions of this stabilizing apparatus, particularly the SSC tendon, may lead to an instability of the LHBT, resulting in the insidious onset of pain at the anterior region of the shoulder, often radiating down the anterior arm over the biceps brachii muscle belly. The symptoms are often exacerbated at night or with overhead activity, arm rotation or lifting.

Ultrasonography (US) and magnetic resonance imaging (MRI) are primary imaging modalities for assessing the soft tissue around the shoulder, including the LHBT and its anchoring structures. US has several strengths over MRI, including a lower cost, higher spatial resolution, dynamic assessment for LHBT instability, interaction with patients during scanning and an easy comparison with the contralateral side [6]. However, US of the musculoskeletal system is highly operator-dependent and requires appropriate training and experience. In this article, US anatomy, scanning techniques and common pathologic conditions encountered whilst evaluating the LHBT anchoring structures are reviewed in order to minimize the limitations of US.

## 2. General Consideration of US

As part of a comprehensive evaluation of the shoulder, when scanning the LHBT and its anchoring apparatus, a linear high frequency (preferably > 12 MHz) transducer should be used. To avoid an anisotropic artifact, the ultrasound beam should be as perpendicular to the target as possible, especially when scanning organized parallel tendon or ligament fibers. The patient should be seated on a revolving stool opposite the examiner. Before making a final diagnosis, all target structures should be evaluated on both the short and the long axis.

## 3. Anatomy and US Technique

### 3.1. Rotator Interval, CHL and SGHL

Before entering the bicipital groove, the LHBT courses through the rotator interval; it is surrounded and stabilized by the CHL and SGHL, which form a sling-like band anchoring the LHBT proximal to the bicipital groove. The US interrogation begins with the evaluation of the rotator interval, which is a triangular space representing a defect of the rotator cuff between the anterior border of the SSP tendon superiorly and the superior border of the SSC inferiorly [2]. The rotator interval includes the LHBT, CHL and SGHL. Originating from the lateral aspect of the coracoid process, the CHL crosses over the LHBT and then bifurcates into a medial and lateral band, inserting at a lesser tuberosity and a greater tuberosity, respectively. The SGHL is a focally thickened band in the glenohumeral joint capsule, which arises from the superior glenoid tubercle, blending with medial band of the CHL that surrounds the medial and inferior aspects of the LHBT and inserts onto the lesser tuberosity [7].

For the evaluation of the CHL and SGHL with US, the patient should be positioned with the hand resting palm up on the thigh and the elbow flexed [8,9,10], leading to the maximum tension of the interval (Figure 1A). The CHL is visualized as a homogeneous echogenic band over the LHBT, measuring 2–3 mm in thickness. As the medial band of the CHL merges with the SGHL, these two ligaments are difficult to distinguish by US [2,10]. The SGHL and medial CHL ligament complex is visualized as a homogeneous echogenic band medial and inferior to the LHBT (Figure 1B).

### 3.2. Bicipital Groove Morphology and Cross-Sectional Area

After the evaluation of the rotator interval, the transducer is moved downward to the bicipital groove. The forearm of the patient is supinated and placed on the ipsilateral thigh. The bicipital groove is visualized in the transverse plane as a concave and highly echogenic line on the US (Figure 2). Theoretically, the osseous dimensions of the bicipital groove contributes to the stability of the LHBT; several studies based on MRI suggest the presence of a spur on the bicipital groove [11], a larger opening angle, a smaller medial angle and a shallower depth are predisposing factors for biceps tendon instability [12]. With ultrasound, however, these dimensions or the cross-sectional area are not related to the stability or other pathologies of the LHBT [13,14]. The bony morphology of the bicipital groove demonstrated by US has a limited value in the diagnosis of various pathologies related to the LHBT. The soft tissue factors above the bicipital groove are obviously more important for maintaining the LHBT stability.

### 3.3. SSC and SSP Tendons

The SSC is a large triangular muscle that fills the subscapular fossa and inserts into the lesser tubercle, acting as an adductor and internal rotator of the arm. A few superficial fibers of the SSC tendon overlay the bicipital groove and reach the greater tuberosity, forming the transverse humeral ligament and merging with the CHL. The SSP originates from the suprascapular fossa, passing under the acromion and above the glenohumeral joint. The SSP muscle assists in not only the stabilization of the shoulder joint but also the abduction of the arm. The SSC and SSP tendons insert onto the lesser and greater tuberosities, respectively, blending with the CHL; they are referred to as the “biceps pulley” [7] whereas the coalescence of the distal SGHL and the medial band of the CHL are called the “reflection pulley” [7] (Figure 3). The CHL, SGHL and fibers from the SSC and SSP tendon merge to form the functional unit that envelopes the LHBT, stabilizing the LHBT and shoulder joint.

To locate the SSC tendon, the hand of the patient should remain palm up on the lap and the arm should be externally rotated with the elbow as close to the body as possible (Figure 4A), bringing the SSC tendon into a more anterior position [15,16]. For the visualizing of the SSP tendon, the hand of the patient should be placed on the buttock with the elbow flexed. In this position, known as the Crass position (Figure 4B), the SSP tendon is placed under stress. On US, the SSC tendon is seen at its insertion site on the lesser tuberosity and extends to the bicipital groove as a uniform hyperechoic fibrillar pattern (Figure 4C). On the transverse images of the SSC tendon, individual echogenic tendon slips of the SSC tendon can be seen due to its multipennate architecture (Figure 4D). Similar to the SSC tendon, the SSP tendon appears to be hyperechoic and fibrillar and lays directly on the humerus (Figure 4E). On the transverse plane, the SSP tendon is 2.0–2.5 cm wide (measured anterior to posterior), immediately posterior to the biceps tendon [15,16] (Figure 4F).

### 3.4. Tendon of PM

The PM has a broad origin from the medial half of the clavicle (clavicular head), sternum and 2nd–6th costal cartilages (sternal heads). The muscle fibers fuse to form a common tendon, which travels laterally and runs anterior to the biceps and coracobrachialis muscle belly. It finally inserts into the lateral lip of the bicipital groove [17,18]. The common tendon of the PM has a characteristic U shape with anterior and posterior layers that are inferiorly continuous [19]. The PM is a strong adductor and internal rotator; its function is complimentary to the SSP. 

When scanning the tendon of the PM, the arm of the patient should be externally rotated in the same posture as scanning the SSC tendon. The transducer should be moved down along the bicipital groove from the level of the SSC tendon to locate the tendon of the PM, which is visualized as an echogenic linear structure superficial to the muscle belly of the biceps and attached onto the lateral lip of the bicipital groove [18] (Figure 5).

### 3.5. Tendons of LD and TM

The LD is a large fan-shaped muscle. It has a broad origin from the spinous processes of the six lower thoracic vertebrae as well as the lower ribs and the iliac crest. The muscle fibers converge and superolaterally move with a narrow insertion on the floor of the bicipital groove of the humerus [20,21]. Compared with the LD, the TM is a smaller rectangular muscle that runs deep and is superior to the LD. It originates from the posterior aspect of the inferior corner of the scapula and medially inserts to the bicipital groove, spiraling around the TM insertion [22,23]. A few of the fibers of the TM and LD tendons form the floor of the bicipital groove. Both the TM and LD perform the function of adduction as well as the extension and internal rotation of the shoulder.

The evaluation of the TM and LD tendons should be performed in a transverse plane by locating the medial lip of the bicipital groove. With the arm of the patient maximally externally rotated, the two tendons should be brought into a more anterior position and the transducer slightly moved medially from the level of the PM tendon. Both the tendons have a typical hyperechoic fibrillar appearance on US, deep to the tendon of PM (Figure 6). The LD and TM muscle bellies lie deep in the coracobrachialis and medial to the tendon.

## 4. Pathologic Conditions

### 4.1. Biceps Pulley Lesions

Pulley lesions can be caused by degenerative changes, an acute injury, chronic repetitive stress or tears in the anterior SSP tendon and superior SSC tendon [7,24] due to their close anatomical insertion relationships. Biceps pulley injuries, which are also referred to as “hidden lesions”, are difficult to diagnose even during open or arthroscopic examinations [25]. Identifying abnormalities of the biceps pulley may not be possible with US. However, certain imaging findings may indicate possible biceps pulley lesions. These include a dislocated biceps tendon, abnormalities of the superior border of the SSC tendon and a chondral print on the humeral head [26] (Figure 7). Although all three signs, including a chondral print, as indicators of a pulley lesion are not very specific, these findings may assist the radiologist in identifying an existing “hidden” abnormality of the biceps pulley, leading to a treatment decision. Zappia et al. investigated the presence of subchondral discontinuities (erosions) of the humeral head at the level of the biceps pulley in two perpendicular planes. They found that US correlated well with the arthroscopy [26]. Therefore, this indirect sign may be beneficial when only the pulley system is torn but the LHB is stable and the SSC tendon is normal.

### 4.2. Adhesive Capsulitis (AC)

Adhesive capsulitis (AC), or frozen shoulder, is a self-limiting disease characterized by a painful restriction of motion that worsens at night. The pathology of the disease is idiopathic synovial inflammation with fibrosis and a decreased compliance of the joint capsule [27]. A shoulder arthrography has previously been the standard imaging study for diagnosing AC. A normal shoulder joint can be easily distended to 14 mL; however patients with AC suffer from decreased capsular distension with a joint volume less than 10 mL, pain after an injection of less than 10 mL of contrast material and the marked loss of the normal axillary fold on the shoulder arthrography [28,29]. US has been increasingly used in the diagnosis of AC. Several US findings have been described, including CHL thickening [9,30], axillary recess capsule thickening [31,32,33] and rotator interval abnormalities [8,9] (Figure 8). A decreased capsular volume results in fluid flowing into the sheath of the LHBT; however, AC is not related to other pathologies of the LHBT. In the institution of the authors, the measurement of the axillary recess capsule thickness is a standard protocol in shoulder examinations and a thickness greater than 4 mm is used to diagnose AC, similar to previous studies [31,32]. Although the diagnosis may be based on the clinical history and a physical examination, US still plays a significant role in confirming the diagnosis of AC and guiding intra-articular corticosteroid injections for the rapid improvement of the pain and range of movement [27,34].

### 4.3. Chronic Pathology of the SSC and SSP Tendons

Chronic pathologies of the SSC and SSP tendons are useful predictors of LHBT pathologies, particularly tendinopathy [11,35,36,37]. Patients with SSC tendon tears are approximately six times more likely to develop a severe grade LHB tendon disorder than those without [11]. A possible explanation for this phenomenon is than chronic rotator cuff tears result in superior humeral head migration relative to the glenoid fossa. The LHBT, a functional humeral head depressor, is subject to an overuse injury [38]. In addition, the LHBT passes through the anterior–superior portion of the rotator cuff tendons, which is the most prevalent site for tears. The chronic pathology of the two tendons would interrupt the stabilizing mechanism of the LHBT. Medial instability is associated with an SSC tear whereas posterolateral instability is associated with an SSP tear [39,40]. Therefore, when there are findings of tears or tendinopathy of the SSC and SSP tendons, radiologists and surgeons should be concerned about the condition of the LHBT.

The US diagnosis of SSC and SSP tendon tears has been well-studied. For full thickness tears, a discontinuity or gap within the tendon filled with anechoic or hypoechoic fluid is observed. In cases of a complete tear with a retraction, the deltoid muscle directly articulates on the humeral head without the visualization of the tendon (Figure 9). For partial thickness tears—which involve only part of the tendon depth—focal flattening, the concavity of the bursal tendon surface or undulation of the tendon contour are observed. When well-trained radiologists and high-resolution transducers are available, US has a comparable accuracy with MRI for detecting rotator cuff tears. However, MRI is superior in surgical planning for larger tears [41,42,43].

### 4.4. PM Tendon Tears

The PM is at risk during any activity in which the arm is extended and externally rotated whilst under a maximal contraction [44]. The most common activity leading to a PM rupture is a bench press exercise [45]. The PM tendon is the anchoring apparatus of the LHBT at its lower part and is the most frequently injured (59%) followed by injuries to the musculotendinous junction (24%) [46]. The US findings of PM tendon tears include disruption and the absence and retraction of the tendon and muscle fibers as well as heterogeneous hematomas [17] (Figure 10). With a partial tear (grade II injury) only involving the posterior layer of the PM tendon, the LHBT will be in place; however, the anterior displacement of the LHBT occurs with a complete tear (grade III injury). Thus, complete tear injuries should be surgically managed with suturing or bone-tunneling techniques [46].

### 4.5. Injuries to the LD and TM

Injuries to the LD and TM are particularly rare and seen most frequently in high-level overhead-throwing athletes [20,21,22]. Injuries to the two muscles and their tendons have been postulated to be caused by an eccentric or a supraphysiological concentric contraction during a throwing motion [47,48]. There are no pathognomonic symptoms or physical exam findings associated with injuries to the LD and TM. Therefore, imaging plays an important role in confirming the diagnosis as well as demonstrating the extent of the injury in order to guide treatment. Partial tendon tears or intramuscular injuries may successfully be treated non-surgically whereas complete tendon ruptures require surgical repair [21]. A tendon injury may be associated with an injury to the adjacent structures such as the rotator cuff or the PM, thus secondarily affecting the LHBT. The most commonly encountered injury pattern is muscle belly strains [22]. Published literature regarding a US diagnosis is sparse. US can demonstrate the discontinuity of the tendon as well as the degree of retraction from the humeral insertion. In cases of an intramuscular injury, the US findings range from an increased echogenicity and swelling to a disrupted fibrillar pattern of the muscle with anechoic to hypoechoic clefts [49] (Figure 11).

## 5. Conclusions

US is a rapid, low-cost and accurate modality for evaluating the anchoring apparatus of the LHBT. Various lesions of these structures can lead to LHBT disorders such as tendinopathy or subluxation. The keys to a successful examination of the LHBT anchoring apparatus include understanding the anatomy, scanning the structures in both the long and the short axis, eliminating anisotropic artifacts and then evaluating the disease. US is a valuable and efficient tool in evaluating biceps pulley lesions, AC, chronic pathology of the SSC and SSP tendons, tears in the PM tendon and injuries to the LD and TM.

## Figures and Tables

**Figure 1 diagnostics-12-00659-f001:**
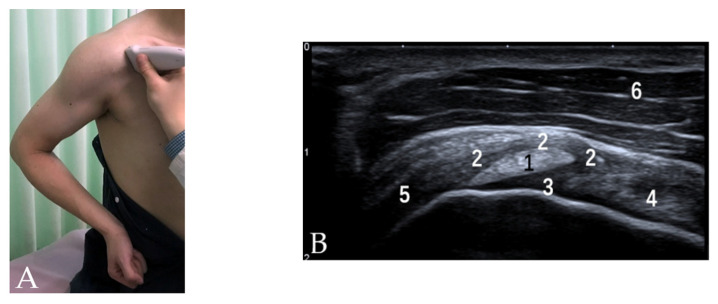
(**A**) Patient position and probe orientation when evaluating rotator interval. (**B**) US image shows rotator interval. 1 = LHBT, 2 = CHL, 3 = SGHL, 4 = tendon of SSC, 5 = tendon of SSP, 6 = deltoid muscle.

**Figure 2 diagnostics-12-00659-f002:**
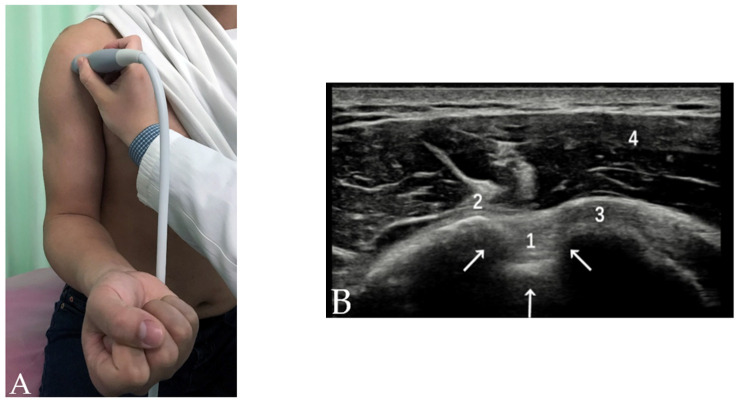
(**A**) Patient position and probe orientation when evaluating LHBT and bicipital groove. (**B**) US image shows the LHBT and bicipital groove. White arrows = bicipital groove, 1 = LHBT, 2 = greater tuberosity, 3 = lesser tuberosity, 4 = deltoid muscle.

**Figure 3 diagnostics-12-00659-f003:**
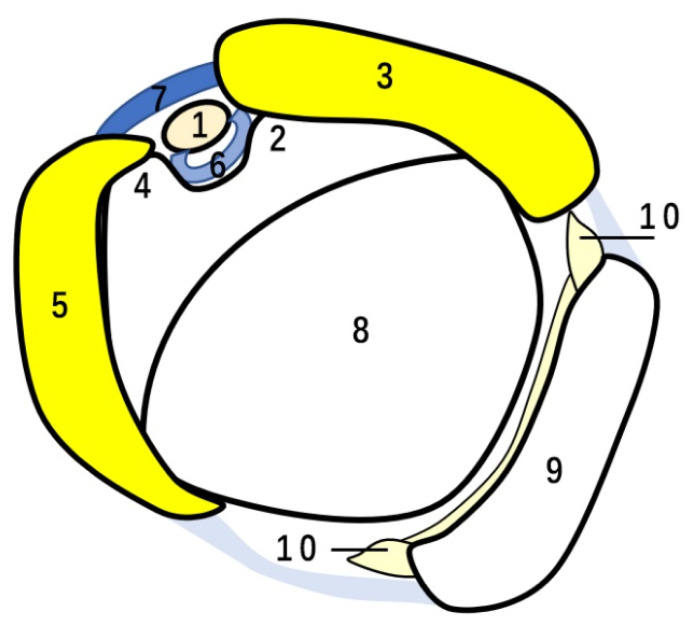
Schematic drawing of biceps and reflection pulley. 1 = LHBT, 2 = lesser tuberosity, 3 = SSC tendon, 4 = greater tuberosity, 5 = SSP tendon, 6 = SGHL, 7 = CHL, 8 = humeral head, 9 = scapular glenoid, 10 = glenoid labrum. In the drawing, 3, 5 and 7 form the biceps pulley and 6 and the medial band of 7 form the reflection pulley.

**Figure 4 diagnostics-12-00659-f004:**
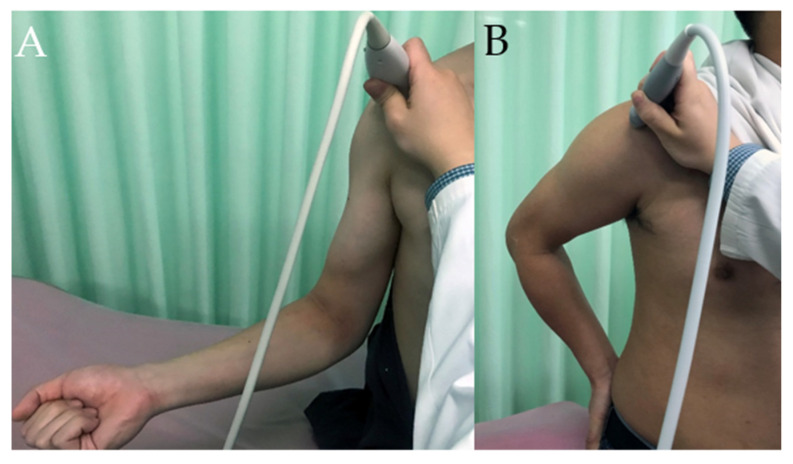
(**A**) Photographs illustrates the patient position and US transducer orientation when evaluating SSC tendon. (**B**) Photograph illustrates the patient position and US transducer orientation when evaluating SSP tendon. (**C**) US image shows long axis of SSC tendon. 1 = SSC tendon, 2 = lesser tuberosity, 3 = deltoid muscle, 4 = subdeltoid bursae. (**D**) US image shows short axis of SSC tendon. (**E**) US image shows long axis of SSP tendon. (**F**) US image shows short axis of SSP tendon. 1 = SSP tendon, 2 = greater tuberosity, 3 = humeral head, 4 = deltoid muscle. Arrow = anatomical neck of humerus.

**Figure 5 diagnostics-12-00659-f005:**
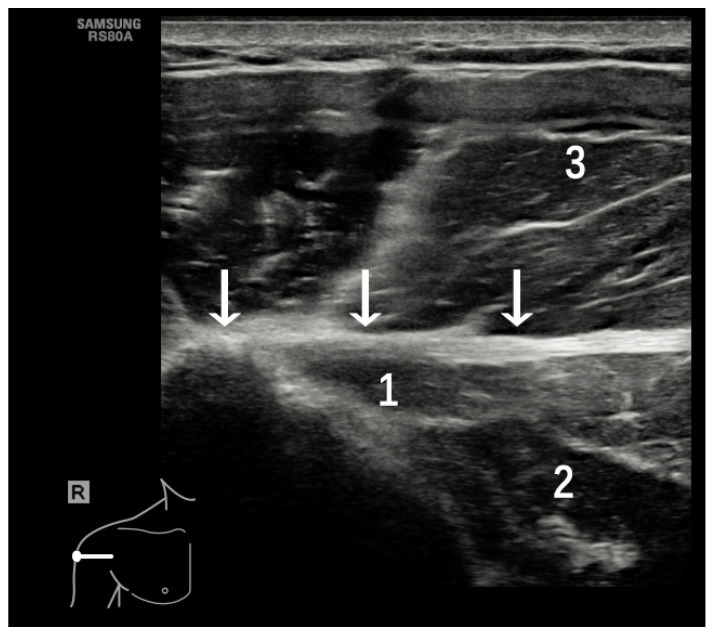
Transverse US image shows an echogenic linear PM tendon (white arrows), which attaches to the lateral lip aspect of the of the bicipital groove. 1 = biceps muscle, 2 = coracobrachialis muscle, 3 = deltoid muscle.

**Figure 6 diagnostics-12-00659-f006:**
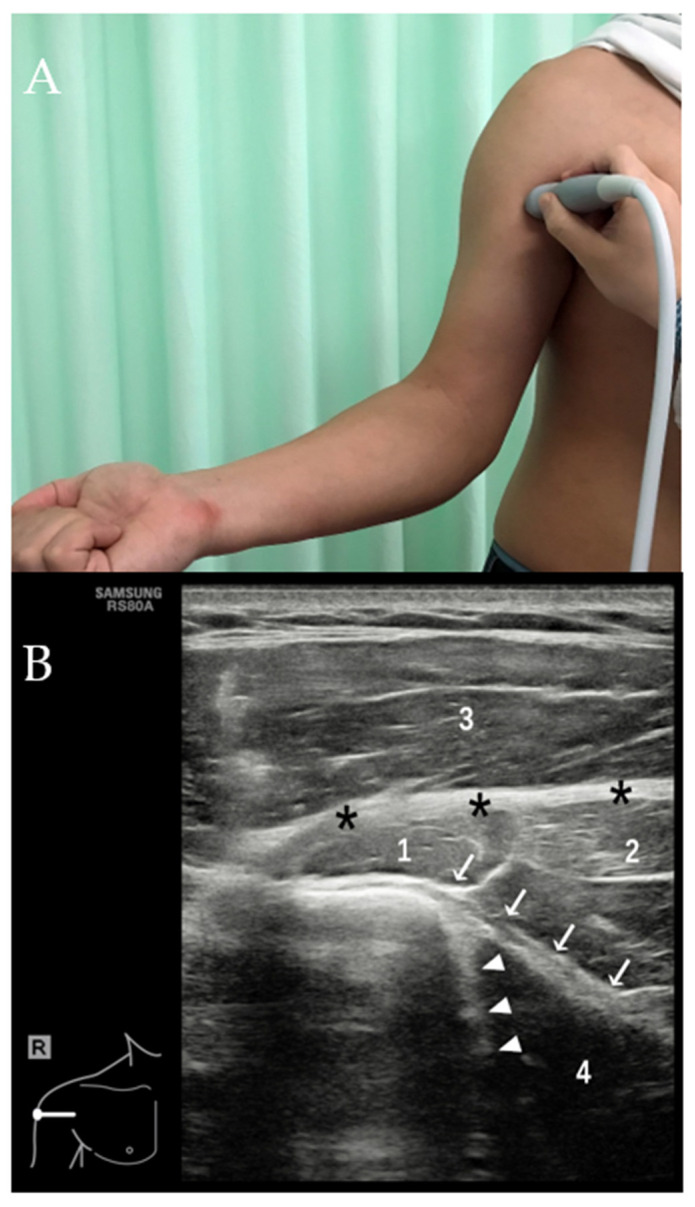
With the arm of the patient maximally externally rotated (**A**), transverse US image (**B**) shows tendons of the LD and TM as echogenic linear structures (white arrows indicate LD tendon and white arrow head indicates TM tendon). Black asterisk = PM tendon, 1 = biceps muscle, 2 = coracobrachialis muscle, 3 = deltoid muscle, 4 = LD muscle. Note that the muscle belly of the LD is significantly hypoechoic due to anisotropic artifact.

**Figure 7 diagnostics-12-00659-f007:**
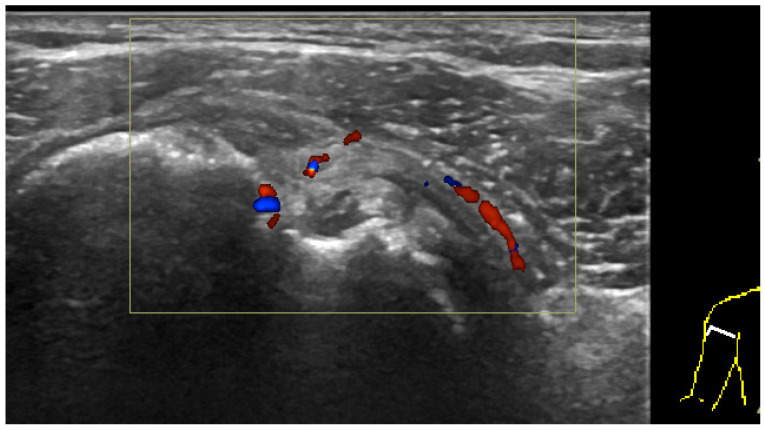
The patient was a 50-year-old female, complaining of right shoulder pain and restricted range of movement for 2 months. Transverse US over the bicipital groove demonstrated medially dislocated LHBT, bony irregularities of humerus and swelling of the biceps pulley with hyperemia. A biceps pulley lesion was confirmed with arthroscopic surgery.

**Figure 8 diagnostics-12-00659-f008:**
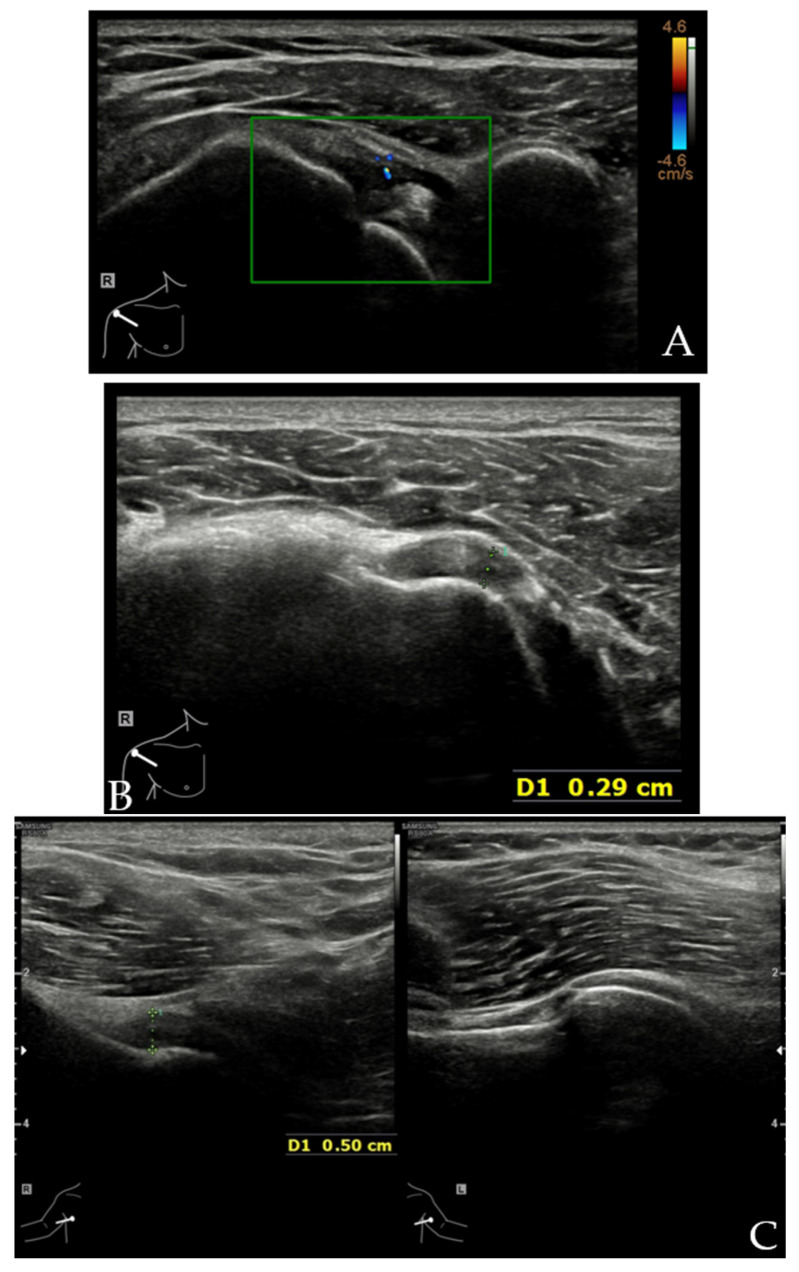
US images of a patient with AC. The patient was a 62-year-old female with painful restricted motion of the right shoulder for 6 months. The clinical diagnosis was AC. (**A**) Transverse image of rotator interval showed increased vascularity. (**B**) Transverse image obtained slightly distally showed increased fluid in the sheath of LHBT resulting from decreased capsular volume of the shoulder joint. (**C**) Axillary recess capsule was significantly thickened (between calipers measuring 0.5 cm) compared with the asymptomatic side. (**D**) Posterior recess capsule was also thickened (between calipers measuring 0.35 cm).

**Figure 9 diagnostics-12-00659-f009:**
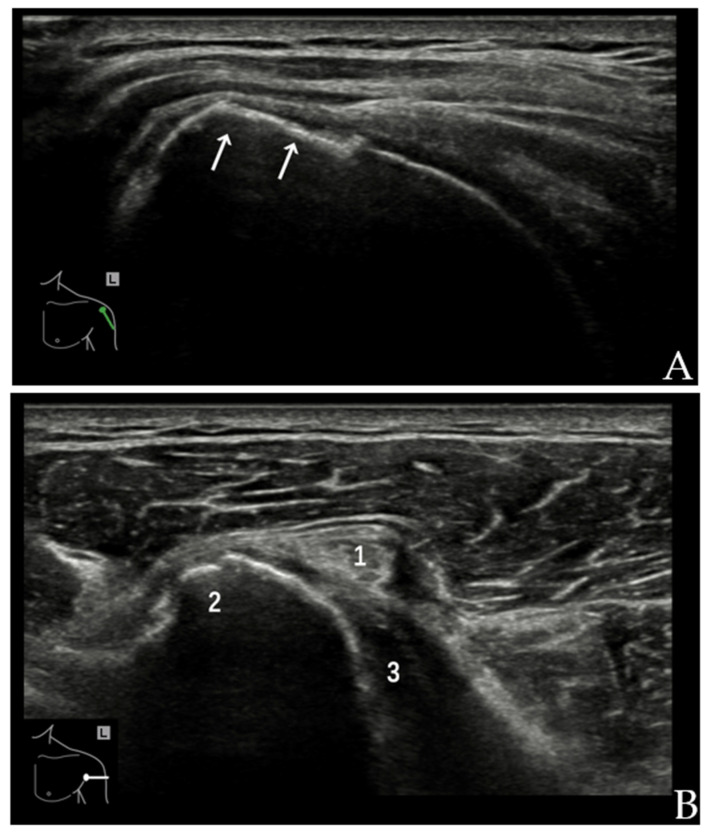
A 63-year-old male patient, complaining of left shoulder pain and palpable snap when the arm was externally rotated. (**A**) Long axis of the SSP tendon showed full thickness tears in the anterior border (white arrow). (**B**) With the arm externally rotated, the short axis of the LHBT showed dislocation of the tendon to the medial side of the lesser tuberosity, superficial to the SSC tendon. (1 = LHBT, 2 = lesser tuberosity, 3 = SSC tendon).

**Figure 10 diagnostics-12-00659-f010:**
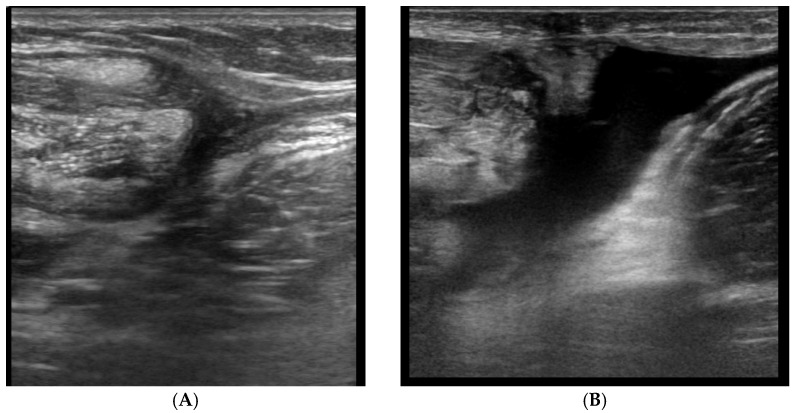
PM tendon tears in 2 patients. (**A**) 64-year-old male patient, complaining of focal pain and weakness with arm adduction after swinging from a high bar. US showed the absence of the PM tendon and retraction of PM muscle fibers. (**B**) 25-year-old male patient, complaining of immediate pain after a bench press exercise. US showed retracted PM muscle stump and surrounding anechoic fluid.

**Figure 11 diagnostics-12-00659-f011:**
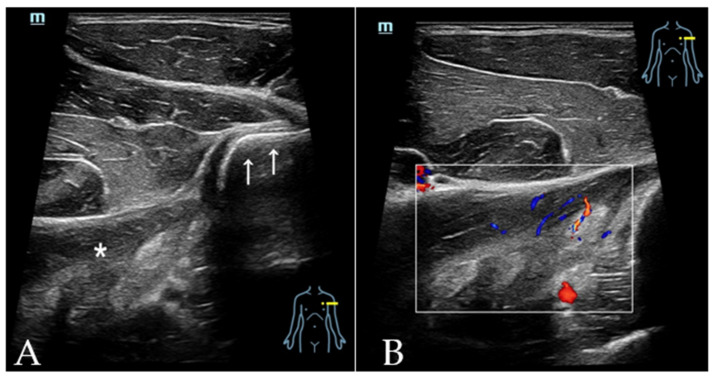
LD muscle belly strains in a 34-year-old professional bodybuilder. He presented with acute and focal pain in the back of the left shoulder after injury. (**A**) Transverse image of the LD tendon (white arrows) showed swelling and disrupted fibrillar pattern of the muscle belly (white asterisk). (**B**) The swelling in the LD muscle belly showed increased vascularity. As the tendon of LD was intact, conservative treatment was recommended.

## Data Availability

Not applicable.

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
