# Peer review of "Anchoring Apparatus of Long Head of the Biceps Tendon: Ultrasonographic Anatomy and Pathologic Conditions"

_diagnostics, 2022, doi:10.3390/diagnostics12030659_

Round 1

Reviewer 1 Report

This study describes the anchoring apparatus of the log head of the biceps muscle and its pathological conditions with ultrasound musculoskeletal ultrasonography (MSKUS) as a diagnostic tool. The paper educates the reader on the apparatus, anatomical components involved, and finally, elucidates the pathologies with MSKUS techniques and findings. It is a well-written paper with reader-friendly language. There are some minor issues that should be fixed before publication.

1- Title, I would suggest not using acronyms in the title.

2- Abbreviations should be defined when used for the first time. Please check the text and figures legend for this issue.

3- Please check the references for the journal format (e.g. Ref. 14)

Author Response

Point 1: 1- Title, I would suggest not using acronyms in the title.

Response 1: Thank you for pointing out, the title has been revised.

Point 2: Abbreviations should be defined when used for the first time. Please check the text and figures legend for this issue.

Response 2: Abbreviations in the text and figure ligend have been checked.

Point 3: Please check the references for the journal format (e.g. Ref. 14)

Response 3: References for the journal format have been checked and revised by using Endnote.

Reviewer 2 Report

An overview of US examination of stabilizers of LHBT is given. In a clear, concise and systematic manner relevant anatomy, US technique and possible pathological findings are discussed, making it useful as a quick search reference for radiology residents and/or junior radiologists. Although numerous, very well written publications already exist about shoulder US, I think the report can still have its place in literature due to its focus. Another strenght of the manuscript is the US image quality, demonstrating the US signs and anatomy nicely. 

However, I have some (minor) remarks

  • I don't feel figure 1 is of significant added value. Although done before, I think that schematic drawings of relevant anatomy would be more contributing to the readability, perhaps incorporated in figures 2-5/6. Maybe separate figure with "biceps/reflection pulley" 
  • minor punctuation issues, mainly spaces, should be checked
  • dedicated x-ray for assessment of the groove seems to be outdated
  • chondral print in general as indicator for pulley lesion is not very specific. 
  • fig 3A: arm position should be displayed as well (like the other images)
  • fig 4b: not exactly the Crass position demonstrated (correctly described in the text)
  • please mark the images with a, b, c, etc..
  • I would suggest reformatting the images in a better way
  • fig 4f demonstrated, with no description
  • fig 6: you can consider an additional image demonstrating the scanning position of the arm

Author Response

Point 1: I don't feel figure 1 is of significant added value. Although done before, I think that schematic drawings of relevant anatomy would be more contributing to the readability, perhaps incorporated in figures 2-5/6. Maybe separate figure with "biceps/reflection pulley"

Response 1: We have deleted figure 1 from the manusciprt and schematic drawings of biceps/reflection pulley have been added.

Point 2: Minor punctuation issues, mainly spaces, should be checked.

Response 2: Punctuation issues have been checked and revised.

Point 3: Dedicated x-ray for assessment of the groove seems to be outdated.

Response 3: Thank you for pointing out, this part, together with previous reference 11 have been deleted from 3.2.

Point 4: Chondral print in general as indicator for pulley lesion is not very specific.

Response 4: Thank you for pointing out. We have revised the manuscript in 4.1.

Point 5: Fig 3A: arm position should be displayed as well (like the other images).

fig 4b: not exactly the Crass position demonstrated (correctly described in the text).

please mark the images with a, b, c, etc.

I would suggest reformatting the images in a better way.

fig 4f demonstrated, with no description.

fig 6: you can consider an additional image demonstrating the scanning position of the arm.

Response 4: Arm position for Fig 3A has been added and Fig 4B is replaced. Images with a, b, c, etc. have been marked. The description of Fig 4F has been added. Arm position for Fig 6 has been added.